# Insulin sensitivity in mesolimbic pathways predicts and improves with weight loss in older dieters

Lena J Tiedemann[1], Sebastian M Meyhöfer[2,3], Paul Francke[1], Judith Beck[1], Christian Büchel[1], Stefanie Brassen[1]*

[1]Department of Systems Neuroscience, University Medical Center Hamburg-Eppendorf, Hamburg, Germany; [2]Institute for Endocrinology & Diabetes, University of Lübeck, Lübeck, Germany; [3]German Center for Diabetes Research (DZD), Ingolstädter Landstraße, Germany

**Abstract** Central insulin is critically involved in the regulation of hedonic feeding. Insulin resistance in overweight has recently been shown to reduce the inhibitory function of insulin in the human brain. How this relates to effective weight management is unclear, especially in older people, who are highly vulnerable to hyperinsulinemia and in whom neural target systems of insulin action undergo age-related changes. Here, 50 overweight, non-diabetic older adults participated in a double-blind, placebo-controlled, pharmacological functional magnetic resonance imaging study before and after randomization to a 3-month caloric restriction or active waiting group. Our data show that treatment outcome in dieters can be predicted by baseline measures of individual intranasal insulin (INI) inhibition of value signals in the ventral tegmental area related to sweet food liking as well as, independently, by peripheral insulin sensitivity. At follow-up, both INI inhibition of hedonic value signals in the nucleus accumbens and peripheral insulin sensitivity improved with weight loss. These data highlight the critical role of central insulin function in mesolimbic systems for weight management in humans and directly demonstrate that neural insulin function can be improved by weight loss even in older age, which may be essential for preventing metabolic disorders in later life.

*For correspondence:
sbrassen@uke.de

## Editor's evaluation

This is a strong translationally relevant study on the importance of insulin and the mesolimbic response to feeding and attempts at weight loss. It will be of great interest to not only neuroscientists but those who study metabolism and nutrition.

## Introduction

The prevalence of overweight and obesity rises dramatically with age and is associated with increased morbidity and reduced quality of life (*Kalyani et al., 2017*). Hyperinsulinemia and insulin resistance are potential causes and consequences of obesity. Both are negatively affected by age and both play a pivotal role for the development of type 2 diabetes and other age-related diseases (*Palmer and Kirkland, 2016*). The mechanisms and directions of these interactions are under debate. For instance, it has long been assumed that insulin resistance in aging precedes the development of hyperinsulinemia, while recent data suggest a reverse direction (*Janssen, 2021*). Moreover, there is evidence from animal and human research of an weight-independent effect of aging on insulin sensitivity (*Ehrhardt et al., 2019*; *Petersen et al., 2003*).

Besides the importance of peripheral insulin for the glycemic control in the body, recent findings also highlight the role of insulin action in the brain for the metabolic and hedonic control of food intake (*Kullmann et al., 2020a*). Findings in rodents and humans indicate that, apart from signaling in hypothalamic neurocircuits regulating energy homeostasis, central insulin mediates non-homeostatic feeding for pleasure by signaling within mesolimbic reward circuits (*Davis et al., 2010*; *Murray et al., 2014*; *Tiedemann et al., 2017*). Specifically, insulin action in the ventral tegmental area (VTA) reduces hedonic feeding in rodents (*Labouèbe et al., 2013*; *Mebel et al., 2012*) and decreases hedonic value signals in the VTA and nucleus accumbens (NAc) in lean subjects (*Tiedemann et al., 2017*). On the other hand, insulin-mediated long-term depression of VTA dopamine neurons is reduced in hyperinsulinemia (*Liu et al., 2013*) and aberrant insulin action in VTA–NAc pathways has been observed in insulin-resistant participants (*Tiedemann et al., 2017*).

There is consensus that the improvement of hyperinsulinemia and insulin resistance, as achieved by caloric restriction (CR), is key in the prevention and treatment of obesity, type 2 diabetes, or cardiovascular diseases in aging (*Janssen, 2021*; *Ryan, 2000*). Evidence from human studies for such effects, however, are sparse, especially when it comes to central nervous insulin signaling. High insulin sensitivity in MEG theta activity was predictive for long-term weight management in 15 young adults (*Kullmann et al., 2020b*) suggesting the critical role of central insulin action for future feeding regulation and as a major target of treatment intervention. Whether such intervention can modulate brain insulin sensitivity in older age is particularly questionable given that changes in function and distribution of adipose tissue can trigger metabolic alterations such as hyperinsulinemia (*Palmer and Kirkland, 2016*; *Tchkonia et al., 2010*) and relevant brain circuits for central insulin action like the dopaminergic mesolimbic pathway undergo age-related changes (*Karrer et al., 2017*).

In the current longitudinal study, we investigated the role of peripheral and central insulin resistance regarding their predictive value for dietary success in older adults and whether both can be modified by weight changes. Fifty older (>55 years) overweight, non-diabetic individuals were randomly assigned to a 3-month, CR intervention or an active waiting group (WG). Before and after the intervention, overnight fasted participants took part in a crossover, placebo-controlled, double-blind

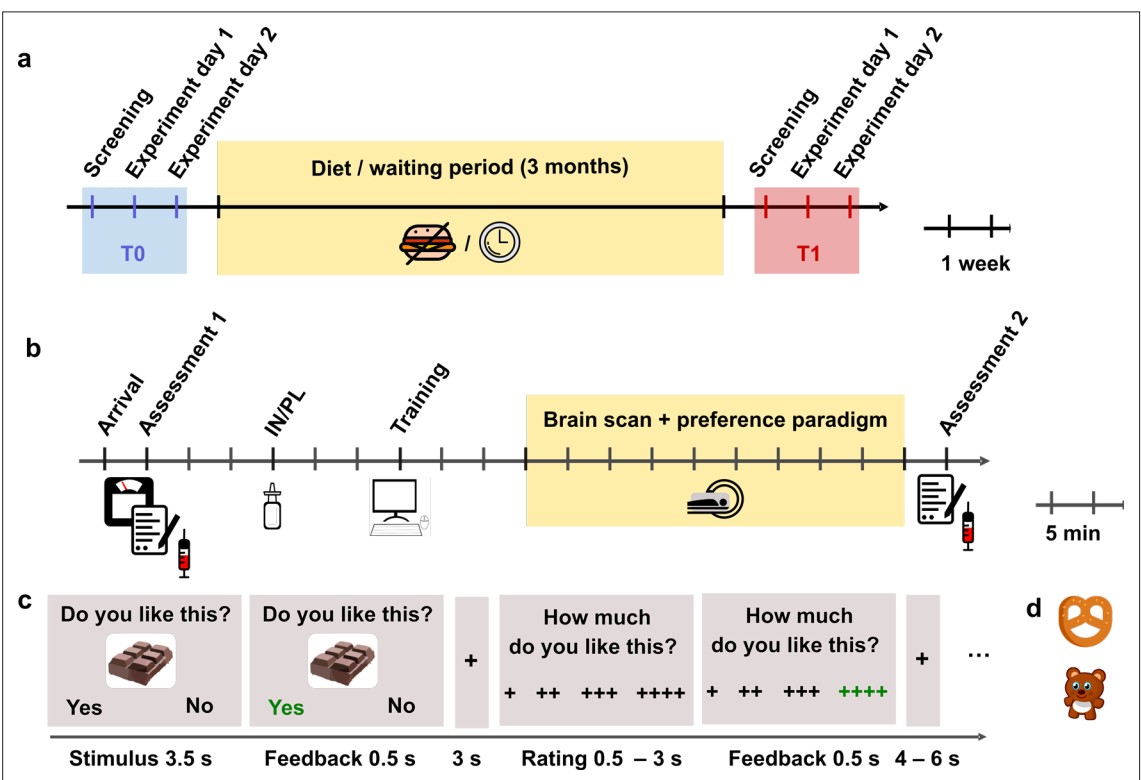

**Figure 1.** Outline of the study design and experimental task. (**a**) Timeline of the longitudinal design. Each participant attended four fMRI sessions, two each before and after the 3-month intervention interval. (**b**) Protocol of the experimental MRI sessions. (**c**) Timing of the fMRI paradigm. Example of a high sugar food trial. (**d**) Examples of low-sweet food and non-food items.

pharmacological fMRI examination in which they rated the palatability of high and low sugar food pictures and the attractiveness of non-food items (control) after receiving intranasal insulin (INI) or placebo. Fasting c-peptides and blood glucose were assessed to calculate peripheral insulin sensitivity. We tested several predictions: (1) successful weight loss can be predicted by peripheral and central insulin sensitivity, the latter indicated by an insulinergic inhibition of mesolimbic responses to hedonic food stimuli at baseline (*Tiedemann et al., 2017*), and (2) both, peripheral and central insulin sensitivity improve with successful weight loss at follow-up. Moreover, we explored the common and distinct impact of both markers on weight (changes) in older age.

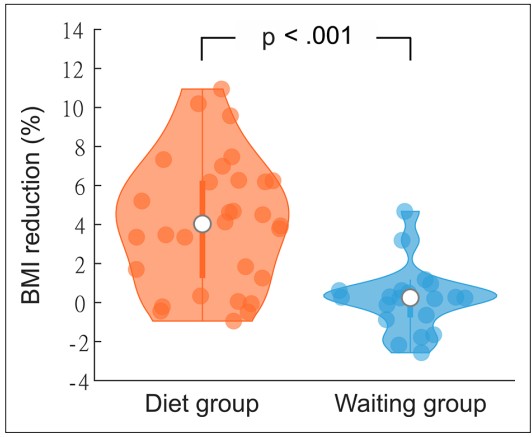

**Figure 2.** Percentage body mass index (BMI) change after 3 months in the diet (N = 30) and the waiting (N = 20) group. Violin plots show individual data, median, interquartile range, and 1.5× interquartile range.

## Results

Fifty overweight and obese older adults (age: 63.7 ± 5.9 years, range 55–78 years; body mass index [BMI]: 32.7 ± 4.3 kg/m², range 25.8–32.4 kg/m²; 20 men) with an explicit wish to lose weight participated in this study. Of these, 30 randomly selected participants underwent a 3-month CR diet (diet group, DG, 14 men), while 20 participants were randomly assigned to a 3-month active WG (6 men). Before (T0) and after (T1) the intervention phase, we assessed anthropometrics and blood measures. Normal HbA1C values confirm the exclusion of manifest diabetes in overweight and obese participants who are at risk for T2D but in whom elevated insulin release may still compensate for reduced insulin sensitivity (mean HOMA-2: 2.2 ± 0.08, range 1.2–3.5). All participants underwent a double-blind, randomized, placebo-controlled fMRI paradigm on food and non-food liking combined with an INI application before and after the intervention phase (*Figure 1*). Thus, each participant attended a total of four scanning sessions. This longitudinal, within-subject design allowed us to evaluate peripheral and food-related central insulin action linked to overweight and weight loss in older adults.

**Table 1.** Sample characteristics at baseline and follow-up.

| | DG (N = 30) | | | WG (N = 20) | | | |
| --- | --- | --- | --- | --- | --- | --- | --- |
| | T0 | T1 | p time | T0 | T1 | p time | p time × group |
| BMI (kg/m²) | 32.1 (0.7) | 30.8 (0.6) | *** | 32.8 (1.1) | 32.8 (1.2) | N.S. | *** |
| Waist (cm) | 103.7 (1.8) | 98.1 (1.9) | *** | 103.1 (2.7) | 99.7 (2.8) | N.S. | N.S. |
| Bodyfat | 37.4 (1.3) | 36.7 (1.4) | N.S. | 39.5 (1.6) | 39.0 (1.7) | N.S. | N.S. |
| Blood | | | | | | | |
| HOMA-2 | 2.1 (0.1) | 1.9 (0.1) | * | 2.3 (0.2) | 2.3 (0.1) | N.S. | N.S. |
| Glucose (mmol/l) | 5.5 (0.1) | 5.4 (0.1) | N.S.[1] | 5.7 (0.1) | 5.8 (0.1) | N.S.[1] | N.S.[2] |
| Insulin (pmol/l) | 78.8 (4.0) | 70.3 (4.4) | + | 95.5 (8.7) | 94.2 (7.7) | N.S. | N.S. |
| C-peptide (nmol/l) | 0.9 (0.03) | 0.8 (0.04) | * | 1.1 (0.06) | 1.0 (0.05) | N.S. | N.S. |
| HbA1C | 5.4 (0.03) | 5.4 (0.04) | N.S. | 5.5 (0.07) | 5.5 (0.07) | N.S. | N.S. |

***p < 0.001, *p < 0.05, +p < 0.10, s.e.m. in parantheses

DG, diet group; WG, waiting group; PL, placebo; IN, insulin; T0, baseline; T1, follow-up; [1] Wilcoxon-rank Test; [2] Mann-Whitney-U-Test.
BMI = body mass index. HOMA-2 = c-peptide-based Homeostatic Model Assessment for Insulin Resistance. N.S = not significant.

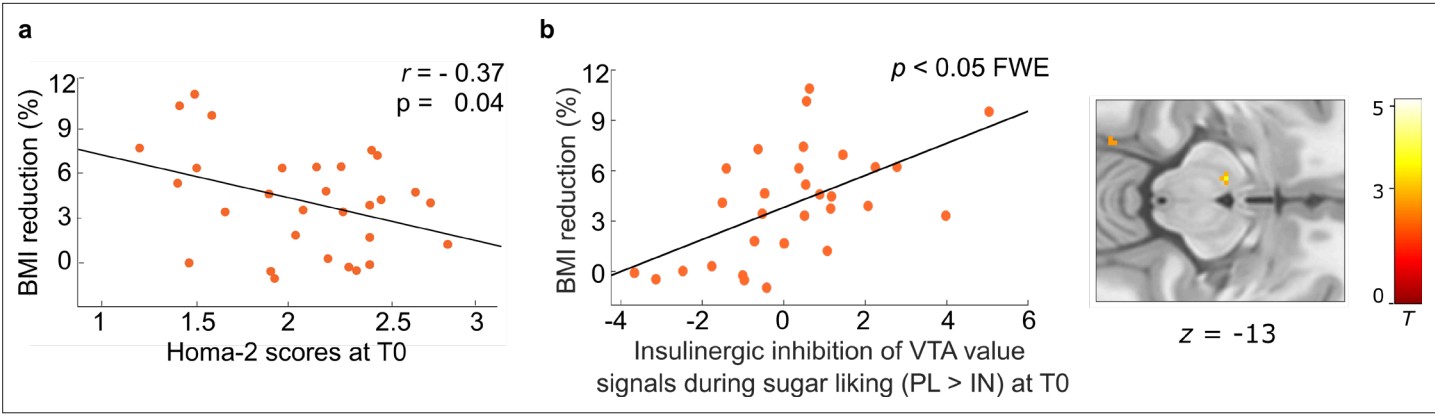

**Figure 3.** Predictors of subsequent weight loss in dieters. (**a**) Higher peripheral insulin sensitivity measured via Homa-2 scores at baseline (T0) was related to higher percentage body mass index (BMI) reduction in dieters (N = 30) at follow-up (T1). (**b**) Correlation between insulin effects in the ventral tegmental area (VTA) at baseline and subsequent weight loss in dieters. Individual blood oxygenation level-dependent (BOLD) signals were extracted from the peak voxel in the VTA resulting from the parametric contrast $PL_{HS>LS} > IN_{HS>LS}$ within dieters, p < 0.05 FWE corrected for bilateral VTA mask.

## CR significantly reduces weight in dieters

Glucose, insulin, and c-peptide levels were assessed on all four study days in the morning after an overnight fast of at least 10 hr. Fasting glucose levels confirmed fasting state in all participants. Groups were well balanced regarding gender, age, overnight fasting times, and days between sessions (all p > 0.28). At T0, BMI, weight, bodyfat, waist, and HOMA-2 did not differ between groups (all p > 0.06, *Table 1*).

After 3 months (mean 96 ± 10 days, no group differences: p = 0.91), follow-up measurements showed a significant mean weight loss compared to baseline of 3.61 kg (±3.06, $T_{(29)}$ = 6.47, p < 0.001, d = 1.18) in the DG, reflecting on average a 4% loss of baseline bodyweight and BMI, respectively ($T_{(29)}$ = 6.66, p < 0.001, d = 1.22). Twenty-one dieters (70%) lost more than 2 kg (range: 2.5–9.6 kg), only one dieter gained more than 0.4 kg (3.1 kg). In the WG, mean weight change was 0.07 kg (±1.5 kg). Sixteen participants of the WG were able to maintain their weight ±2 kg. Two gained weight (2.3 and 2.7 kg), two lost weight of 2.3 and 4 kg, respectively. Accordingly, percentage BMI change differed significantly between groups ($BMI_{\%change}$: both $T_{(48)}$ = 5.45, p < 0.001, d = 1.39; *Figure 2*, *Appendix 1—figure 1*).

## Baseline peripheral insulin sensitivity predicts dietary success

Fasted serum c-peptide and plasma glucose levels were used for the calculation of an effective measure of peripheral insulin resistance (HOMA-2, *Levy et al., 1998*; https://www.dtu.ox.ac.uk/homacalculator/index.php). To test the predictive value of baseline insulin resistance in the periphery for dietary success after 3 months, we correlated individual HOMA-2 scores assessed on the placebo session from T0 with $BMI_{\%change}$ in dieters and found a significant correlation (r = −0.37; p = 0.046, n = 30, Pearson's correlation). That is, higher insulin sensitivity at baseline predicted more subsequent weight loss in dieters (*Figure 3a*). Control analyses showed no such correlation in the WG and no correlation was observed between $BMI_{\%change}$ and baseline BMI (all p > 0.16).

## Insulinergic inhibition of sweet food liking at baseline predicts dietary success

At T0, after an overnight fast of at least 10 hr (day 1: 12.5 ± 1.6 hr; day 2: 12.2 ± 1.6 hr, no group differences), all participants underwent a 2-day fMRI scanning procedure, separated by at least 1 week (9.0 ± 3.4 days) that was combined with 160 IU INI or placebo in a double-blind, randomized crossover design (*Figure 1*). Fasting time and hunger ratings did not differ between groups (all p > 0.32; *Appendix 1—table 1*).

In the scanner, participants were asked to rate the overall preference for high (HS) and low sugar (LS) food and non-food items with yes (~'I like this') or no (~'I do not like this') by button press, which was followed by a four-point rating scale where they were asked to provide a detailed rating, indicating

how much they liked or disliked each item. Stimuli were presented in pseudo-randomized order. Parametric values were derived from transferring the general and the four-point rating into a single scale ranging from 1 ('not at all') to 8 ('very much') (validation study of all four sets, *Appendix 1—table 2*).

Placebo and insulin sessions did not differ across individuals with respect to prescan insulin, c-peptide (all p > 0.30, *n* = 50, *t*-test), glucose, hunger ratings, and time fasted (all p > 0.24, *n* = 50, Wilcoxon test), nor were there any group × session differences in these parameters (all p > 0.11, $n_{DG}$ = 30, $n_{WG}$ = 20, *t*-test, Mann–Whitney *U*-test). Similarly, changes in pre- compared with post-hunger ratings, as well as levels of glucose, did not differ between the placebo and the insulin session across and between groups (all p > 0.32; Wilcoxon test, Mann–Whitney *U*-test). As expected, plasma insulin levels across all participants decreased over time ($F_{(1,48)}$ = 38.79, p < 0.001, $\eta^2$ = 0.45, repeated measures analysis of variance [rmANOVA]); there was a lower decrease at the insulin day across participants ($F_{(1,48)}$ = 4.08, p = 0.049, $\eta^2$ = 0.08, rmANOVA) but not within single groups or as group interaction (p > 0.83; *Appendix 1—table 1*).

As expected, in the T0 placebo session, food items were liked significantly more than non-food items on the categorical (yes/no) and parametric (cumulated ratings 1–8) level (all p < 0.001). Preference values for HS and LS food did not differ across or between groups (all p > 0.15). C-peptide-based insulin sensitivity was correlated with HS liking, in a way that higher insulin sensitivity was related to lower HS liking (*r* = 0.38; p = 0.006; *n* = 50, Pearson's correlation, *Appendix 1—figure 2*). There was no relationship of HOMA-2 scores with LS liking (p > 0.16).

We then investigated the effects of INI on preference values at T0. Analyses across and between groups yielded no significant differences between the placebo and the insulin session, neither for food > non-food nor for HS > LS food items (all p > 0.14, rmANOVA). There was no interaction between insulin effects and insulin sensitivity as assessed by HOMA-2. To investigate the predictive value of individual differences in insulin effects on future weight changes we added $BMI_{\%change}$ as a covariate into the analyses (rmANCOVA). While there was no interaction with insulin effects on general food versus non-food values, analysis on sugar-specific values (HS > LS) demonstrated a significant two-way interaction session × $BMI_{\%change}$ ($F_{(1,48)}$ = 6.24; p = 0.016, $\eta^2$ = 0.12, rmANCOVA). To further explore this finding, the analysis of insulin effects was limited to participants with a minimum BMI reduction of 1% (*n* = 25). In this subsample, insulin decreased sugar preference (i.e., percentage of sweet foods in preferred foods) at baseline significantly ($T_{(24)}$ = 2.10; p = 0.046, n = 25, d = 0.42, *t*-test) and this effect could not be explained by BMI or HOMA-2 (all p > 0.12).

## Midbrain insulin effects during sugar liking predict weight loss in dieters

To examine the neural mechanisms of how insulin influenced the brain's mesocorticolimbic reward circuitry, we analyzed blood oxygenation level-dependent (BOLD) activity measured during the preference task using a two-level random effects model. As expected from our previous study (*Tiedemann et al., 2017*), the analysis of differences in BOLD responses to food compared to non-food items in the placebo session at T0 yielded highly significant activations across all participants in a network of reward-related brain regions including the bilateral insula, medial OFC, and amygdala (*Figure 4a*). Also in line with our previous findings, regions that encode the subjective value of items, that is, regions that show a positive correlation between the amplitude of the BOLD response and subjective preference values, comprised regions of the brain's valuation network including the vmPFC and NAc (*Figure 4b*). BOLD signals in these regions did not differ between groups. Furthermore, valuation of HS compared to LS food items evoked significantly stronger correlations between BOLD signal and preference values in the ACC/vmPFC (*Figure 4c*), the right caudate nucleus and thalamus (all p < 0.05 FWE corrected) and as trend in the right NAc (9, 10, −7; FWE = 0.09, *Figure 4c*).

We then investigated the effects of INI on these value signals. Here, in line with behavioral findings, analyses across and between groups yielded no significant changes of neural value signals for both general food items and HS versus LS items (see uncorrected results in *Appendix 1—figure 3*). There was also no relation between HOMA-2 and neural insulin effects across individuals or within the DG. Following up on the behavioral findings, we next analyzed whether individual insulin effects on neural signals during HS compared to LS food valuation predicted subsequent weight loss in the DG. Simple regression analysis including $BMI_{\%change}$ as a covariate yielded insulin-induced signal changes in the left VTA to predict subsequent weight loss across all participants as well as within subjects from the DG

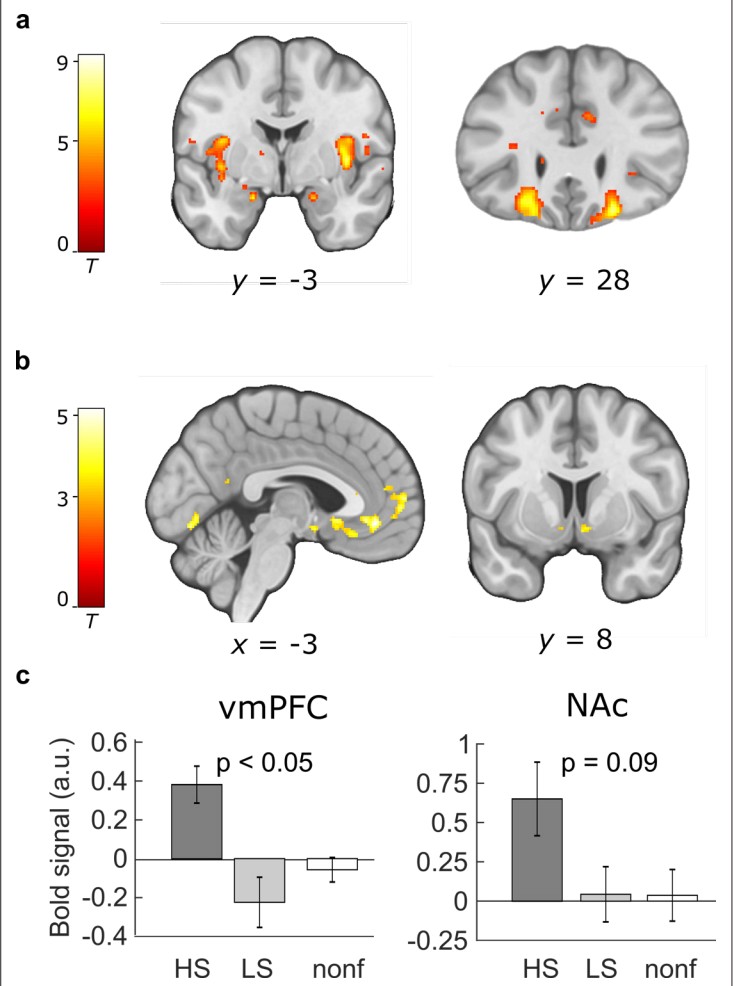

**Figure 4.** Paradigm-induced activation patterns during baseline placebo. (**a**) Categorical effect of food stimulus presentation. Greater activity in the insula, amygdala, and orbitofrontal cortex was observed in the food compared to the non-food condition across both groups (N = 50; included contrast images: food > non-food). (**b**) Neural representation of preference values (parametric analysis). Regions in which the correlation with preference values was significant across participants included the ventromedial prefrontal cortex (vmPFC) and the bilateral nucleus accumbens (NAc) (included contrast images: all food × liking). (**c**) Sugar-specific blood oxygenation level-dependent (BOLD) signals in the vmPFC and right NAc. Bar plots show means and standard error of the mean (SEM) of contrast estimates extracted from peak voxels from the comparison HS > LS (included contrast images: HS × liking > LS × liking). All peaks and displayed p values are p < 0.05 FWE corrected. Activations are overlaid on a custom template (display threshold p < 0.005 uncorrected).

alone ($-8$, $-13$, $-13$, p < 0.05 FWE corrected; *Figure 3b*). Results in the left VTA were still significant when including age and BMI as covariates in the analysis (p < 0.05 FWE corrected). This indicates that participants in whom INI reduced the HS-specific valuation signal in the midbrain at T0 are more likely to benefit from CR by weight loss as assessed at T1.

## Baseline central and peripheral insulin sensitivity make independent contributions to the prediction of dietary success

To assess the incremental predictive value of baseline peripheral and central insulin sensitivity for weight changes after 3 months of intervention, we then performed a multiple regression analysis using HOMA-2 scores and the extracted BOLD signal from the contrast $PL_{HS>LS} > IN_{HS>LS}$ in the VTA to predict $BMI_{\%change}$. Within participants from the DG, this model turned out to be highly significant ($F_{(2,27)} = 10.07$; adjusted $R^2 = 0.39$; p < 0.001), with both predictor variables explaining substantial variability (VTA-Bold: $\beta = 0.54$; $T = 3.70$; p < 0.001; HOMA-2: $\beta = -0.35$; $T = 2.41$; p = 0.023, p < 0.025

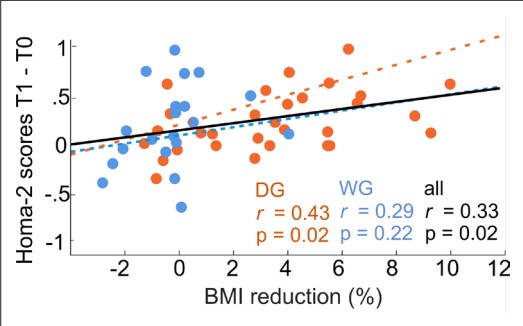

**Figure 5.** Weight loss is related to improvements in peripheral insulin sensitivity. Within dieters (N = 30) and across participants (N = 50) a higher percentage of body mass index (BMI) changes was correlated with an increase of insulin sensitivity as measured via the Homa-2 scores.

Bonferroni corrected). Inclusion of BMI in a control model demonstrate these effects not to be driven by baseline bodyweight (*Appendix 1—table 3*). This indicates, that peripheral and central insulin sensitivity at baseline have an independent positive impact on subsequent weight loss in overweight older dieters.

## Improvement of peripheral insulin sensitivity is related to increased insulin effects on NAc HS value signals after weight loss

We finally investigated metabolic and neurobehavioral changes due to successful weight loss. Within participants from the DG, HOMA-2 scores were significantly improved at follow-up ($T_{(29)}$ = 2.33; p = 0.027, d = 0.43). Moreover, the improvement in HOMA-2 scores was directly correlated with successful weight change within dieters (r = 0.43; p = 0.017; N = 30, Pearson's correlation) and across all participants (r = 0.33; p = 0.020; N = 50, Pearson's correlation, *Figure 5*).

We next tested whether successful weight loss also improved central insulin sensitivity as assessed with our pharmacological fMRI design (for characteristics of the T1 fMRI sessions see *Appendix 1—table 1*, *Appendix 1—table 4*). In behavior, participants from the DG showed a significantly reduced sweet food preference (i.e., percentage of sweet foods in preferred foods) under insulin compared to placebo at follow-up ($T_{(29)}$ = 2.59; p = 0.015, d = 0.47) that tended to be stronger compared to the WG ($T_{(49)}$ = 1.80; p = 0.08) and to baseline ($T_{(29)}$ = 1.67; p = 0.11) (*Figure 6a*).

On the neural level, behavioral insulin effects in the DG at follow-up were reflected by a stronger reduction of sugar-specific value signals in the NAc under insulin in the DG compared to the WG (peak right: 10, 8, −7, p = 0.028 FWE corrected and peak left: −10, 12, −8, p = 0.043 FWE corrected, t-test). The effect in the right NAc was also significantly stronger when directly comparing follow-up to baseline valuation responses between groups (peak: 10, 8, −6, p = 0.019 FWE corrected; two-sample t-test). Exploration of extracted BOLD signals (*Figure 6b*) indicate that this effect was at least partly driven by an opposite effect in the WG, that is, sweet food signals in the NAc relatively increased

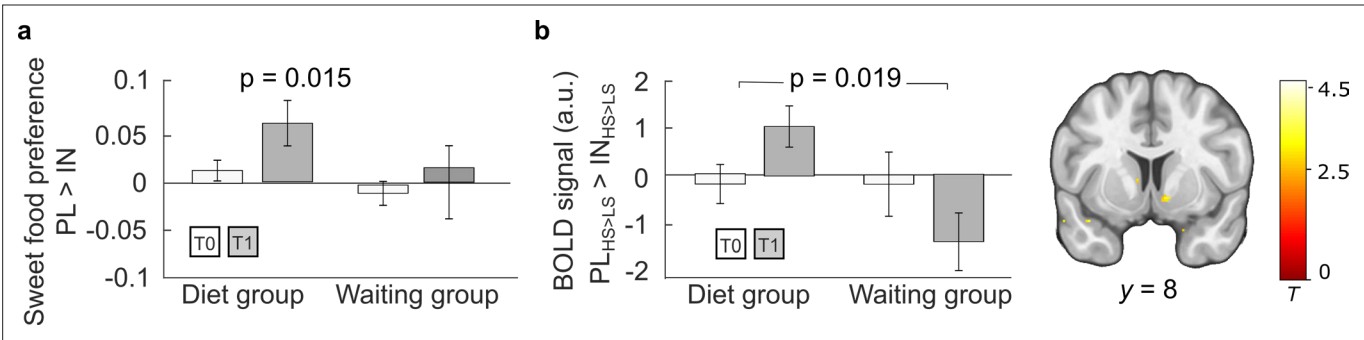

**Figure 6.** Central insulin effects on behavior and brain activity before and after 3 months. (**a**) Behavioral insulin effects on sweet food preference. While there was no insulin effect observed at baseline T0 in both groups, the percentage of preferred sweet food items decreased significantly under insulin compared to placebo at follow-up in dieters. (**b**) General linear modeling of sweet versus non-sweet value signals under insulin compared to placebo revealed a significantly stronger signal decrease in the diet group (N = 30) compared to the waiting group (N = 20) at follow-up (T1) compared to baseline (T0) in the right nucleus accumbens (NAc) included contrast images: T1 [PL$_{HS>LS}$ > IN$_{HS>LS}$] > T0 [PL$_{HS>LS}$ > IN$_{HS>LS}$]. Bar plots show group means and standard error of the mean (SEM) of mean contrast estimates extracted from significant peak voxel. p < 0.05 FWE corrected for bilateral NAc mask. Activations are overlaid on a custom template (display threshold p < 0.005 uncorrected).

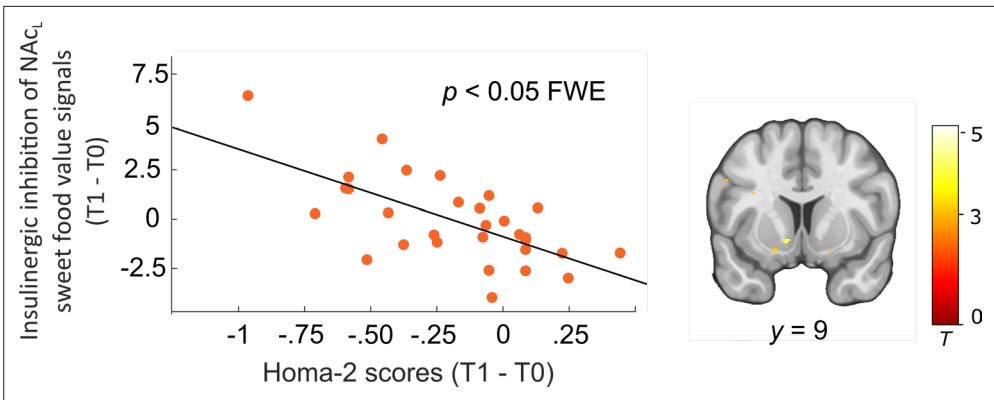

**Figure 7.** Interaction between peripheral and central insulin changes in dieters (N = 30). Improved (T1 > T0) insulinergic inhibition of sweet food value signals in the left nucleus accumbens (NAc) correlated with improved peripheral insulin sensitivity. Individual blood oxygenation level-dependent (BOLD) signals were extracted from the peak voxel in the right NAc resulting from simple regression analysis including the parametric contrast HS > $LS_{PL>IN\_post}$ > HS > $LS_{PL>IN\_pre}$ and HOMA-2 changes (post > pre) as covariate of interest. p < 0.05 FWE corrected for bilateral NAc mask. Activations are overlaid on a custom template (display threshold p < 0.005 uncorrected).

under insulin at T1. There was no significant insulin effect across all participants at T1 (see uncorrected results in *Figure 3b*).

We therefore more directly focused on the DG and explored whether insulin-mediated neural signal changes after weight loss were related to changes in peripheral insulin sensitivity following dietary intervention. To this end, changes in HOMA-2 scores (post > pre) were entered as a covariate in a simple regression of BOLD signals from the contrast HS > $LS_{PL>IN\_post}$ > HS > $LS_{PL>IN\_pre}$ within participants from the DG. Results revealed positive correlations between improvement of insulin sensitivity measured in the blood and increased insulin effects in the NAc (peak left: −10, 10, −7, p = 0.03 FWE corrected; peak right: 10, 10, −7; *Figure 7*). No significant brain correlates for this analysis were found in the WG. Results in the left NAc were still significant when including age and BMI as covariates in the analysis.

## Discussion

Our findings indicate an independent predictive value of peripheral and central insulin sensitivity for dietary success in overweight elderlies and an improvement of both after losing weight. In non-diabetic, overweight, and obese older participants who underwent a 3-month CR, significant weight loss could be predicted by baseline measures of c-peptide-based insulin sensitivity as well as acute insulinergic inhibition of VTA responses to high sugar food items. Both markers of insulin function made an independent contribution to weight loss prediction emphasizing the necessity to take both aspects into account when assessing predictors and consequences of overweight in the elderly. At follow-up, weight loss in dieters was associated with improved peripheral insulin sensitivity which was directly related to a stronger insulinergic inhibition in the NAc during hedonic food valuation. These findings extend work in rodents (*Mebel et al., 2012*) and first studies in humans (*Kullmann et al., 2020b*) about the critical role of insulin sensitivity for future feeding regulation. It is also, to our knowledge, the first study that demonstrates positive effects of CR on central insulin function in humans using a longitudinal within-subject design. The observation of such an effect in older adults is particularly important given age-related metabolic and neural changes and the potential, detrimental consequences of hyperinsulinemia, overweight, and obesity in later life (*Janssen, 2021*).

As was expected from overweight and obese individuals, both peripheral and central markers of insulin sensitivity were low at baseline and both selectively improved with weight loss after CR in dieters. At baseline, participants with higher hyperinsulinemia demonstrated a specifically enhanced sugar preference which fits animal and human data on the critical role of sugar-enriched diets on whole-body insulin functioning (*Macdonald, 2016*). Baseline insulin markers in the blood, however, were not related to behavioral or neural INI responses to HS items. This may be driven by the general

lack of central insulin effects across participants at baseline which fits with data in overweight younger adults (*Tiedemann et al., 2017*) and which might result from an attenuated insulin transport into the cerebrospinal fluid in individuals with reduced whole-body insulin sensitivity (*Heni et al., 2014*). However, there was also no such association in the subgroup of successful dieters in whom there was a significant response to INI already before the intervention. This indicates that, even though peripheral insulin might be a good proxy for central insulin functioning in younger lean adults (*Tiedemann et al., 2017*), pathophysiological changes due to overweight and aging might at least in part independently affect peripheral and central insulin effects. This is further underlined by the independent predictive value of both markers for weight changes after 3 months of CR. The impact of these predictive values could not be explained by body mass, which demonstrates that insulin sensitivity, but not necessarily obesity, is predictive for future weight management. Indeed, there is evidence from animal and human research of an adiposity-independent effect of aging on insulin sensitivity (*Ehrhardt et al., 2019*; *Petersen et al., 2003*), that, for instance, may result from age-related changes in mitochondrial energy metabolism (*Petersen et al., 2003*) or increased systemic inflammation (*Ehrhardt et al., 2019*).

Inhibitory INI effects on VTA value signals to sweet food items were selectively predictive for subsequent success of CR. The VTA plays a central role in the insulinergic modulation of hedonic eating behavior (*Labouèbe et al., 2013*; *Mebel et al., 2012*; *Tiedemann et al., 2017*). Direct administration of insulin into the VTA reduces hedonic feeding and depresses somatodendritic DA in the VTA which has been attributed to the upregulation of the number or function of DA transporter in the VTA (*Mebel et al., 2012*). Connectivity analyses of fMRI data further suggest that INI can suppress food value signals in the mesolimbic pathway by negatively modulating projections from the VTA to the NAc (*Tiedemann et al., 2017*). Insulinergic effects at baseline and follow-up were specifically restricted to HS food stimuli. The palatability of sugar has been linked to DA release in the NAc in rodents (*Hajnal et al., 2004*) and there is evidence for neural adaptations in the NAc in response to excessive sugar intake (*Klenowski et al., 2016*). For instance, higher sugar preference in overweight individuals has been related to stronger white matter connectivity within the VTA–NAc pathway (*Francke et al., 2019*). Our data indicate that insulinergic functionality in this network may be critical for hedonic feeding regulation as the reduction of sugar intake is substantial for the success of a dietary intervention.

A reduced insulinergic functionality of this network in older overweight individuals may not only be the consequence of adiposity (*Mattson and Arumugam, 2018*) but may also result from age-related metabolic and neural changes. Aging is associated with a decrease of cortical insulin concentration, reduced insulin receptor binding ability and reduced insulin transport across the blood–brain barrier (*Cholerton et al., 2011*). Moreover, target systems of metabolic–hedonic networks relevant for insulin action undergo age-related changes (*Mattson and Arumugam, 2018*; *Smith et al., 2020*). The dopamine system, for example, is particularly vulnerable to aging which might lead to functional changes in subcortical reward circuits (*Dreher et al., 2008*; *Karrer et al., 2017*). There is a significant loss of dopaminergic neurons in the basal ganglia including the VTA (*Siddiqi et al., 1999*). Given that insulin acts via glutamatergic synaptic transmission onto VTA DA neurons (*Labouèbe et al., 2013*) this might have direct consequences on the insulinergic suppression of subsequent DA release in mesolimbic regions. The potential negative impact of adiposity and age on described dysfunctions are thereby probably not simply additive. For instance, chronic metabolic morbidities like obesity can further accelerate brain aging (*Mattson and Arumugam, 2018*). A chronic positive energy balance thereby adversely affect brain function (*Beyer et al., 2017*) and structure (*Janowitz et al., 2015*) and is related to many of the cellular and molecular hallmarks of brain aging such as oxidative damage and neuroinflammation (*Mattson and Arumugam, 2018*).

Intriguingly, while there was no association between blood parameters and central insulin action at baseline, weight-change related improvements in peripheral and central insulin sensitivity in our sample of older dieters were directly correlated at follow-up, indicating a common modulator. Moreover, changes in central insulin sensitivity were restricted to an increased inhibition of value signals in the NAc but not the VTA. Thus, one could speculate that weight change specifically normalized adiposity-related dysfunctions while variability due to aging itself were less affected. Improvement in insulin sensitivity and glucose homeostasis is a broadly observed metabolic effect of CR in rodents (*Yu et al., 2019*; *Zhang et al., 2021*) as well as young and older adults (*Fontana and Klein, 2007*;

*Johnson et al., 2016*; *Most and Redman, 2020*; *Most et al., 2017*). The mechanisms behind these effects are not fully understood yet but have been related to significantly increased hepatic insulin clearance (*Bosello et al., 1990*), reduced levels of thioredoxin-interacting protein (TXNIP; *Johnson et al., 2016*), and generally decreased oxidative stress and inflammatory processes (*Fontana and Klein, 2007*). Animal data about CR effects on brain functioning suggest that CR can induce adaptive cellular responses that can enhance neuroplasticity and stress resistance, for example, by the upregulation of neurotrophic factor signaling, suppression of oxidative stress and inflammation, stabilization of neuronal calcium homeostasis, and stimulation of mitochondrial biogenesis (*Mattson, 2012*; *Mattson and Arumugam, 2018*). In addition, recent work in rodents demonstrate improved insulin sensitivity following CR that was associated with enhanced brain monoamine concentrations such as increased DA levels in the striatum (*Portero-Tresserra et al., 2020*). Our data extend these beneficial neural effects of CR in animals to improved central insulin functioning in the human brain. This is particularly intriguing with regard to our non-diabetic sample of elderlies, in whom weight-related brain dysfunction is not only a risk factor for metabolic disorders but also for cognitive decline and neurodegeneration (*Ekblad et al., 2017*; *Janssen, 2021*; *Mattson and Arumugam, 2018*).

We chose a relatively mild dietary intervention that reduced participants' individual caloric intake by 10–15% with a minimal intake set to 1200 kcal per day. This was done to increase compliance and to provide elderlies with a feasible long-term strategy to lose and maintain weight. Accordingly, there was only a mild-to-moderate average weight loss of 4%. Even this mild weight change was related to significant improvement of insulin sensitivity in the periphery and in the brain which underline that adiposity-related dysfunctions in later life are able to normalize. This is especially promising given new evidence for hyperinsulinemia preceding insulin resistance (*Janssen, 2021*) which makes it a key target for early interventions. It is now critical to understand the long-term effects of such changes with a special focus on food intake assuming that long-term effects are probably particularly dependent on prefrontal mediated psychological strategies including self-control during eating decisions (*Hare et al., 2009*; *Phelan et al., 2020*). In conclusion, we provide data demonstrating that peripheral insulin sensitivity as well as central hedonic feeding regulation predict and normalize with dietary success in overweight elderlies. Our results of an independent contribution peripheral and central insulin sensitivity make for successful feeding regulation emphasize the necessity to control for both when treating individuals at risk for metabolic disorders.

## Materials and methods

### Participants

Sixty-four overweight and obese participants (age >55, BMI >25 kg/m²) with an explicit wish to lose weight were recruited for this study. Thirty-eight participants were randomized to the dietary intervention group while 26 were randomly assigned to the WG. Randomization was based on a predefined randomization list (allocation scheme 60:40) and was applied consecutively. Out of these 38 participants from the DG, two did not come back for the follow-up measurement, three individuals showed elevated glucose levels (cutoff ≥126) before at least one scanning session indicating they were not fasted, two showed incomplete task understanding (i.e., always pressed the same button), and one participant had to be excluded due to massive movement artifacts in the scanner. Out of the initial 26 members of the WG, two did not show up for the follow-up measurement, three had substantially increased insulin levels, and from one participant no task behavior could be recorded in the T1 insulin session due to technical issues. This led to a final sample size of 50 complete datasets (55–78 years, $M = 63.7$, standard error [SD] = 5.9, 20 men), 30 derived from the DG and 20 derived from the WG. Mean BMI was 32.4 kg/m² (25.9–43.6, SD = 4.3). Sample characteristics are summarized in *Table 1*.

Sample sizes were based on previous findings on successful CR in older adults (*Witte et al., 2009*). A dropout rate of 25% was considered in our recruitment scheme. Sensitivity measures derived from G*Power 3.1.9 for the final sample sizes indicate our design to be sensitive to detect small ($N = 50$) to medium ($N = 30$) effects in one-sample and paired $t$-tests and large effects in two-sample $t$-tests ($N = 30$, $N = 20$) given an $\alpha$ of 0.05 and $\beta$ of 0.80.

Participants were recruited via newspapers and online announcements. Exclusion criteria were current or previous psychiatric or neurological disorders, chronic and acute physical illness including diabetes, current psychopharmacological medication as well as MR-specific exclusion

criteria. Initial screening as well as all clinical measurements in this study were performed by a physician (P.F. and K.G.). No participant followed any specific diet at the start of the experiment. To exclude systematic confounds during food evaluation, severe food allergies and adherence to a vegan diet constituted further exclusion criteria. All participants had normal or corrected-to-normal visual acuity. The study was approved by the local ethics committee of the Ethical Board, Hamburg, Germany. All participants gave informed consent and were financially compensated for their participation. Additional financial incentives (50€) were provided to participants from the DG for successful weight loss (≥4 kg) and to participants from the WG for keeping their weight stable (weight changes ≤2 kg). The whole study was conducted at the Department of Systems Neuroscience, University Medical Centre Hamburg-Eppendorf. The study has been registered at DRKS (DRKS00028576).

## Experimental protocol

### Baseline (T0)

After successful screening, participants attended two experimental sessions, separated by at least 1 week. On each day, participants arrived in the morning between 7:30 and 10:30 hr after an overnight fast of at least 10 hr. After anthropometric measurements, ratings of feelings of current hunger and collection of blood samples, participants received 160 IU of insulin (Insuman Rapid, 100 IU/ml) or vehicle (0.27% m-Kresol, 1.6% glycerol, and 98.13% water) by intranasal application. Participants received eight puffs per nostril, each puff consisting of 0.1 ml solution containing 10 IU human insulin or 0.1 ml placebo. The order of insulin and placebo was randomized and balanced, and the application was double blind. Before scanning, participants were familiarized with the task during a training session. Participants began the preference paradigm (in the fMRI scanner) 30 min after the nasal spray was applied; this delay was introduced to ensure that the insulin had time to take its full effect (**Born et al., 2002**). After completion of the scans, participants again rated their feeling of hunger and a second set of blood samples was collected (**Figure 1**).

### Intervention

Directly following the second scanning session, participants were randomly assigned to either the DG or the WG following a 60:40 randomization scheme. Participants of the DG received a 12-week professional diet program that consisted of (1) individual nutrition counseling by experienced clinical dieticians, who were blinded to the underlying study hypothesis, and (2) a psychological group intervention (**Appendix 1—figure 4**).

Within individual sessions and based on individual dietary records from the previous 7 days, dieters received an individually planned dietary regimen that reduced each subject's individual caloric intake by 10–15%. Minimal intake was set to 1200 kcal per day. The regimen was based on the 10 guidelines of the German Nutrition Society (https://www.dge.de/). Compliance was assured by three telephone contacts after 2, 6, and 12 weeks during which participants confirmed that they continued to adhere to the diet and had the opportunity to clarify any questions regarding the dietary regimen. After 2 weeks, participants additionally attended a 90-min group session consisting of psychoeducation regarding obesity and T2D, a mindfulness-based eating awareness training and a training of self-monitoring and -control strategies (e.g., use of goal-related eating protocols). In a final counseling, the dietary regimen was reviewed and future eating behavior was discussed.

Participants from the WG were instructed to not change previous eating habits during the 3-month period. In week 6, they attended a 90-min group session which consisted of a psychoeducational unit about stress and stress management as well as a training of progressive muscle relaxation. After finishing all experimental sessions, participants of the WG were offered gratis dietary counseling identical to the one offered to the DG.

### Follow-up (T1)

Three months after the last baseline measurement, participants from both groups repeated the double-blind randomized crossover design from T0, that is they attended another three study days (screening, fMRI + placebo, fMRI + insulin).

## Blood measures

On each scanning day before insulin/placebo application, blood samples were collected, containing the following: 2.7 ml blood in a sodium fluoride for analysis of blood glucose, 7.5 ml blood in a serum tube for analysis of insulin and c-peptide. After completion of MR scans blood sample collection for insulin and glucose analysis was repeated. After 10 min of centrifugation (2800 × $g$ and room temperature), the supernatants of the blood samples were stored at −80°C until further processing. Concentrations of insulin and c-peptide were measured using an electro-chemiluminescence immuno-assay (Roche, ECLIA). Blood glucose was quantified through photometry (Beckman Coulter). We used fasted c-peptide serum levels for the calculation of an effective measure of insulin resistance (HOMA-2; *Levy et al., 1998*, https://www.dtu.ox.ac.uk/homacalculator/index.php). A c-peptide-based index is thought to be a more reliable indicator of insulin secretion that is minimally affected by hepatic insulin clearance, has longer half-life and that is more sensitive to incident T2D (*Jones and Hattersley, 2013*; *Leighton et al., 2017*; *Okura et al., 2018*). Indeed, in a control analysis we observed a significantly higher within-subject variability (coefficient of variance, COV) in prescan insulin levels at T0 as well as T1 compared to c-peptide levels (T0: $T_{(49)}$ = 4.79; p < 0.001; T1: $T_{(49)}$ = 4.47; p < 0.001).

## Statistical analyses

Behavioral and metabolic data processing were conducted using MATLAB (Mathworks, MA) and SPSS 27 (IBM, NY). We report statistical tests from the general linear model framework, including one-sample *t*-tests, two-sample *t*-tests, rmANOVA, multiple regression, and Pearson's correlations. We used the Kolmogorov–Smirnov test to test the null hypothesis that our data come from a normal distribution. In case when data were not normally distributed non-parametric testing was applied using the Wilcoxon test and the Mann–Whitney *U*-test. Statistical significance was assumed based on an alpha value of 0.05. Bonferroni correction was applied on multiple regression coefficients.

## fMRI food-rating paradigm

Four sets of stimuli were randomly presented on the four scanning days. Each one of the four parallel versions consisted of 70 food and 70 non-food color images selected from the internet. All pictures had a size of 400 × 400 pixels and were presented on a white background. Food pictures featured both sweet and savory items. Pictures were specifically selected to cover common high- and low-palatable foods. The four food sets did not differ in sugar content (see *Appendix 1—figure 5* for distribution of sugar content). Non-food pictures, such as trinkets and accessories were chosen to evoke similar degrees of attractiveness. Validation of all four sets was conducted in an independent sample (*n* = 16) and revealed that the four sets did not differ significantly regarding the mean preference ratings (all p > 0.43). Importantly, preference ratings as well as picture saliency for HS and LS stimuli did not differ between sets (*Appendix 1—table 2*).

On each scanning day, food and non-food stimuli were pseudo-randomly presented (not more than three pictures from one category in a row) during three runs; each run lasted ~12 min and runs were separated by a 1-min relaxation break. Every run began with the instructions ('We will soon start with the question: Do you like the presented item or not?') (*Figure 1*).

## MRI data acquisition

All imaging data were acquired on a Siemens PRISMA 3T scanner (Erlangen, Germany) using a 32-channel head coil. Functional data were obtained using a multiband echo-planar imaging sequence. Each volume of the experimental data contained 60 slices (voxel size 1.5 × 1.5 × 1.5 mm) and was oriented 30° steeper than the anterior to posterior commissure (AC–PC) line (repetition time [TR] = 2.26 s, echo time [TE] = 30 ms, flip angle = 80°, field of view [FoV] = 225 mm, multiband mode, number of bands: 2). An additional structural image (magnetization prepared rapid acquisition gradient echo [MPRAGE]) was acquired for functional preprocessing and anatomical overlay (240 slices, voxel size 1 × 1 × 1 mm).

## fMRI data analysis

Structural and functional data were analyzed using SPM12 (Welcome Department of Cognitive Neurology, London, UK) and custom scripts in MATLAB. All functional volumes were corrected for rigid body motion and susceptibility artifacts (realign and unwarp). The individual structural T1 image

was coregistered to the mean functional image generated during realignment. Image diagnostics was performed using visual inspection of image-to-image variability (tsdiffana, https://imaging.mrc-cbu.cam.ac.uk/imaging/DataDiagnostics). The functional images were spatially normalized and smoothed with a 4-mm full-width at half maximum isotropic Gaussian kernel.

A two-level random effects approach utilizing the general linear model as implemented in SPM12 was used for statistical analyses. At the single subject level, onsets of HS, LS, and non-food stimuli presentation were modeled as separate regressors convolving delta functions with a canonical hemodynamic response function. In addition, combined rating scores were entered as parametric modulators of HS, LS, and non-food regressors separately. Onsets of HS and LS were defined based on median splits on sugar content (g/100 g) of the respective stimulus set. Importantly, sugar medians did not differ between sets for placebo/insulin sessions before and after the intervention (all p > 0.25; mean sugar median = 12.8 g sugar/100 g). Data from the placebo and the insulin sessions were defined as single models. In all analyses, we accounted for the expected distribution of errors in the within-subject (dependency) and the between-group factors (unequal variance).

For each subject, contrast images of interest were then entered into second-level group analyses, that is one sample and two-sample *t*-tests. Contrast images of interest comprised onset and parametric regressors for food >non-food and HS > LS from the placebo session at T0 (general paradigm-induced activation) and parametric regressors for HS > LS covering insulin effects at baseline (PL > IN) and compared to follow-up (T1 > T0). We report results corrected for FWE due to multiple comparisons. We conducted this correction at the peak level within small volume of interest (ROI) for which we had an a priori hypothesis or at the whole-brain level. Based on findings in our previous work using the identical pharmacological fMRI setup in younger individuals (*Tiedemann et al., 2017*), we focused on the NAc and the VTA. We applied the identical functional ROIs (4 mm spheres) centered on the bilateralized peak voxels in the NAc (±12, 10, −8) and the VTA (±4, −14, −12) as in this previous work and as identified via meta- analyses conducted on the neurosynth.org platform.

## Acknowledgements

We thank V Ott and M Hallschmid for advice on the INI application setup and K Giesen for help in data acquisition. We gratefully acknowledge funding from the German Research Foundation (DFG, TRR 134, TRR 289 – Project-ID 422744262, BR2877/6-1).

## Additional information

### Competing interests

Christian Büchel: Senior editor, *eLife*. The other authors declare that no competing interests exist.

### Funding

| Funder | Grant reference number | Author |
|---|---|---|
| Deutsche Forschungsgemeinschaft | TRR 134 | Stefanie Brassen<br>Lena J Tiedemann<br>Sebastian M Meyhöfer<br>Paul Francke<br>Judith Beck<br>Christian Büchel |
| Deutsche Forschungsgemeinschaft | TRR 289 - ID 422744262 | Christian Büchel<br>Stefanie Brassen |
| Deutsche Forschungsgemeinschaft | BR2877/6-1 | Stefanie Brassen |

The funders had no role in study design, data collection, and interpretation, or the decision to submit the work for publication.

### Author contributions

Lena J Tiedemann, Data curation, Formal analysis, Investigation, Visualization, Writing - original draft, Project administration; Sebastian M Meyhöfer, Methodology, Writing - original draft; Paul Francke,

Data curation, Investigation, Writing - original draft, Project administration; Judith Beck, Data curation, Investigation, Project administration; Christian Büchel, Conceptualization, Funding acquisition, Writing - original draft; Stefanie Brassen, Conceptualization, Resources, Formal analysis, Supervision, Funding acquisition, Visualization, Methodology, Writing - original draft, Project administration

## Author ORCIDs
Christian Büchel http://orcid.org/0000-0003-1965-906X
Stefanie Brassen http://orcid.org/0000-0002-8884-7593

## Ethics
Informed consent and consent to publish was obtained in all participants. The study was approved by the local ethics committee of the Ethical Board, Hamburg, Germany (Hamburger Ärztekammer, ID: PV4463).

## Decision letter and Author response
Decision letter https://doi.org/10.7554/eLife.76835.sa1
Author response https://doi.org/10.7554/eLife.76835.sa2

# Additional files

## Supplementary files
• Transparent reporting form

## Data availability
All data for analyses and figures in this study are provided at Dryad.

The following dataset was generated:

| Author(s) | Year | Dataset title | Dataset URL | Database and Identifier |
|-----------|------|---------------|-------------|-------------------------|
| Brassen S | 2022 | Data from: Insulin sensitivity in mesolimbic pathways predicts and improves with weight loss in older dieters | http://dx.doi.org/10.5061/dryad.8cz8w9gsn | Dryad Digital Repository, 10.5061/dryad.8cz8w9gsn |

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

## Appendix 1

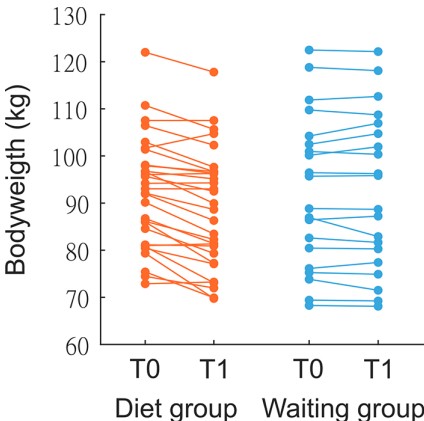

**Appendix 1—figure 1.** Individual weight change in dieters and controls. Weight change in kg after 3-month dietary intervention/waiting phase.

**Appendix 1—table 1.** Fasting duration before each study day and hunger ratings before each MRI scan.

| WG | T0 | | p | T1 | | p | p |
|---|---|---|---|---|---|---|---|
| | PL | IN | (session) | PL | IN | (session) | (session × time) |
| Fasting time (hr) | 12.4 (0.3) | 12.8 (0.4) | N.S. | 12.5 (0.3) | 12.6 (0.4) | N.S. | N.S. |
| Hunger rating | 2.5 (0.6) | 2.2 (0.5) | N.S. | 2.8 (0.5) | 3.1 (0.7) | N.S. | N.S. |

| DG | T0 | | p | T1 | | p | p |
|---|---|---|---|---|---|---|---|
| | PL | IN | (session) | PL | IN | (session) | (session × time) |
| Fasting time (hr) | 12.3 (0.3) | 12.2 (0.3) | N.S. | 12.4 (0.3) | 12.3 (0.3) | N.S. | N.S. |
| Hunger rating | 2.2 (0.4) | 2.3 (0.4) | N.S. | 2.9 (0.5) | 2.8 (0.5) | N.S. | N.S. |

Neither did the fasting times between the last food intake and the beginning of the study day differ between sessions (PL/IN, T0/T1) or groups, nor was there a group x session effect. Before entering the scanner, participants rated their current feelings of hunger on a scale from 0 ("not hungry at all") to 10 ("extremely hungry"). Values did not differ between sessions or groups and there was no group x session interaction. Values indicate means with s.e.m. in parentheses. DG: diet group, WG: waiting group, PL: placebo, IN = insulin, T0: baseline, T1: follow-up. N.S. not significant. [1] Because at least one measure of this variable (across all time-points) was not normally distributed, a Wilcoxon-Rank-Testing was applied.

**Appendix 1—table 2.** Stimuli characteristics.

| | Set 1 | Set 2 | Set 3 | Set 4 | p |
|---|---|---|---|---|---|
| Parametric liking score All food items | 2.86 (0.07) | 2.90 (0.07) | 2.98 (0.06) | 2.96 (0.06) | N.S. |
| HS food items | 3.00 (0.09) | 2.80 (0.10) | 3.03 (0.08) | 2.87 (0.08) | N.S. |
| LS food items | 2.70 (0.09) | 3.00 (0.10) | 2.91 (0.10) | 3.04 (0.10) | N.S. |
| Picture saliency All food items | 0.19 (0.01) | 0.21 (0.01) | 0.18 (0.01) | 0.20 (0.01) | N.S. |
| HS food items | 0.19 (0.01) | 0.21 (0.01) | 0.18 (0.01) | 0.20 (0.01) | N.S. |
| LS food items | 0.20 (0.01) | 0.20 (0.01) | 0.21 (0.01) | 0.20 (0.01) | N.S. |

In a validation study, an independent sample of 16 participants rated the preference of items on a scale from 1 (~ "I do not like this at all") to 4 (~ "I like this very much"). Saliency is calculated based on the Image Signature algorithm, as described by **Hou et al., 2012**. One-way ANOVAs showed that the four sets did not differ in regard to the parametric liking (all P > 0.07) and saliency (all P > 0.43) scores across food items and for HS and LS food items separately. Values indicate means with s.e.m. in parentheses. N.S. not significant.

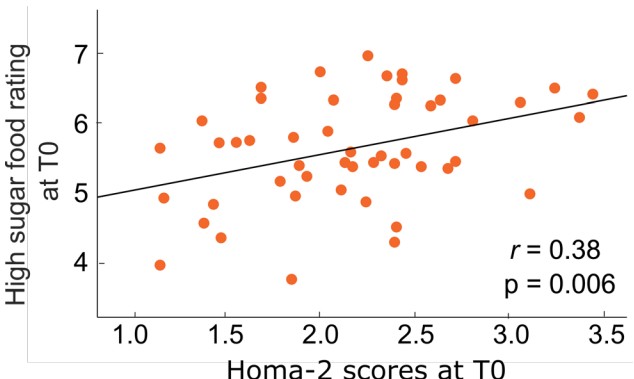

**Appendix 1—figure 2.** Correlation between peripheral insulin sensitivity and sweet food liking. Lower insulin sensitivity as measured via the HOMA-2 score was related to higher sugar liking. No correlation with insulin sensitivity was found for low sugar liking (p > 0.17).

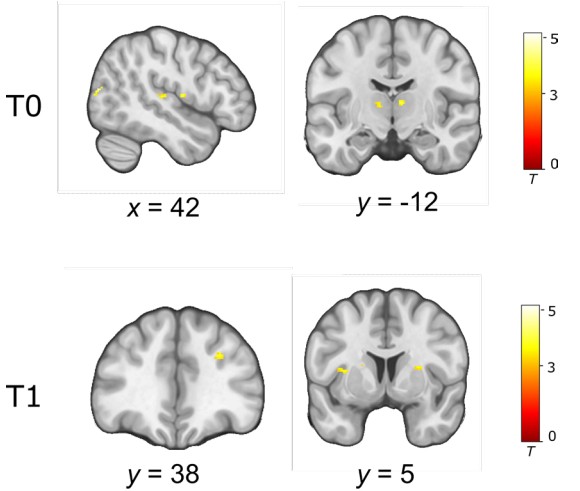

**Appendix 1—figure 3.** Uncorrected whole-brain response to insulin (HS > LS) at T0 and T1 across all participants. Activations are overlaid on a custom template (display threshold p < 0.001, $k = 10$).

**Appendix 1—table 3.** Regression models for the prediction of dietary success.

| | | Predictor variable (standardized $\beta$-coefficients) | | |
|---|---|---|---|---|
| Dependent variable | Model ($R^2$ adjusted) | Insulin sensitivity (HOMA-2) | Insulin effects on VTA signal | BMI |
| Model 1 | | | | |
| BMI change (%) | 0.39*** | −0.35*c | 0.54***c | |
| Model 2 | | | | |
| BMI change (%) | 0.37** | −0.38* | 0.51**c | 0.07 |

***p < 0.001, **p < 0.01, *p < 0.05, csignificant after Bonferroni correction.

**Appendix 1—table 4.** Pre–post blood values at baseline and follow-up.

| DG_T0 | PL | | P | IN | | P | P |
|---|---|---|---|---|---|---|---|
| | Pre | Post | | Pre | Post | | (interaction) |
| **Insulin (pmol/l)** | **78.8 (4.0)** | **62.8 (4.6)** | **0.003** | **80.6 (4.5)** | **72.9 (6.0)** | **0.16** | **N.S.** |
| Glucose (mmol/l) | 5.5 (0.1) | 5.6 (0.1) | N.S. | 5.4 (0.1) | 5.4 (0.1) | N.S. | N.S. |
| **DG_T1** | | | | | | | |
| Insulin (pmol/l) | 70.3 (4.4) | 57.7 (3.8) | 0.001 | 72.9 (4.9) | 62.6 (4.9) | 0.036 | N.S. |
| Glucose (mmol/l) | 5.4 (0.1) | 5.4 (0.1) | N.S.[1] | 5.5 (0.1) | 5.3 (0.1) | 0.02 | N.S.[1] |

| WG_T0 | PL | | P | IN | | P | P |
|---|---|---|---|---|---|---|---|
| | Pre | Post | | Pre | Post | | (interaction) |
| Insulin (pmol/l) | 97. 9 (8.6) | 70.2 (8.1) | 0.001 | 88.6 (7.4) | 70.2 (6.5) | 0.001 | N.S. |
| Glucose (mmol/l) | 5.7 (0.1) | 5.7 (0.1) | N.S. | 5.7 (0.1) | 5.6 (0.1) | N.S. | N.S. |
| **WG_T1** | | | | | | | |
| Insulin (pmol/l) | 97.3 (7.9) | 65.3 (6.0) | 0.001 | 103.0 (8.9) | 80.5 (7.8) | 0.001 | N.S. |
| Glucose (mmol/l) | 5.8 (0.1) | 5.7 (0.1) | N.S. | 5.9 (0.1) | 5.7 (0.1) | 0.02 | N.S. |

Blood samples were sampled after arrival and after completion of the scanning sessions (see **Figure 1b**). There was a significant insulin level x session interaction across participants at T0 (F(1,49) = 4.1; P = .047, rmANOVA) driven by a stronger insulin decrease in the placebo session. This effect was not significant within single groups, in interaction with groups, nor were there any significant session effects at T1 (all P > .30). Values indicate means with s.e.m. in parentheses. DG: diet group, WG: waiting group, PL: placebo, IN = insulin, T0: baseline, T1: follow-up. N.S. not significant. [1] Because that at least one measure of this variable (across all time-points) was not normally distributed, a Wilcoxon-Rank-Testing was applied.

| Eating protocols | Individual nutrition counseling | Follow-up call | Group intervention | Follow-up call | Final interview |
|---|---|---|---|---|---|
| - 1week | week 1 | week 2 | week 3 | week 4 | week 12 |

**Appendix 1—figure 4.** Roadmap dietary intervention.

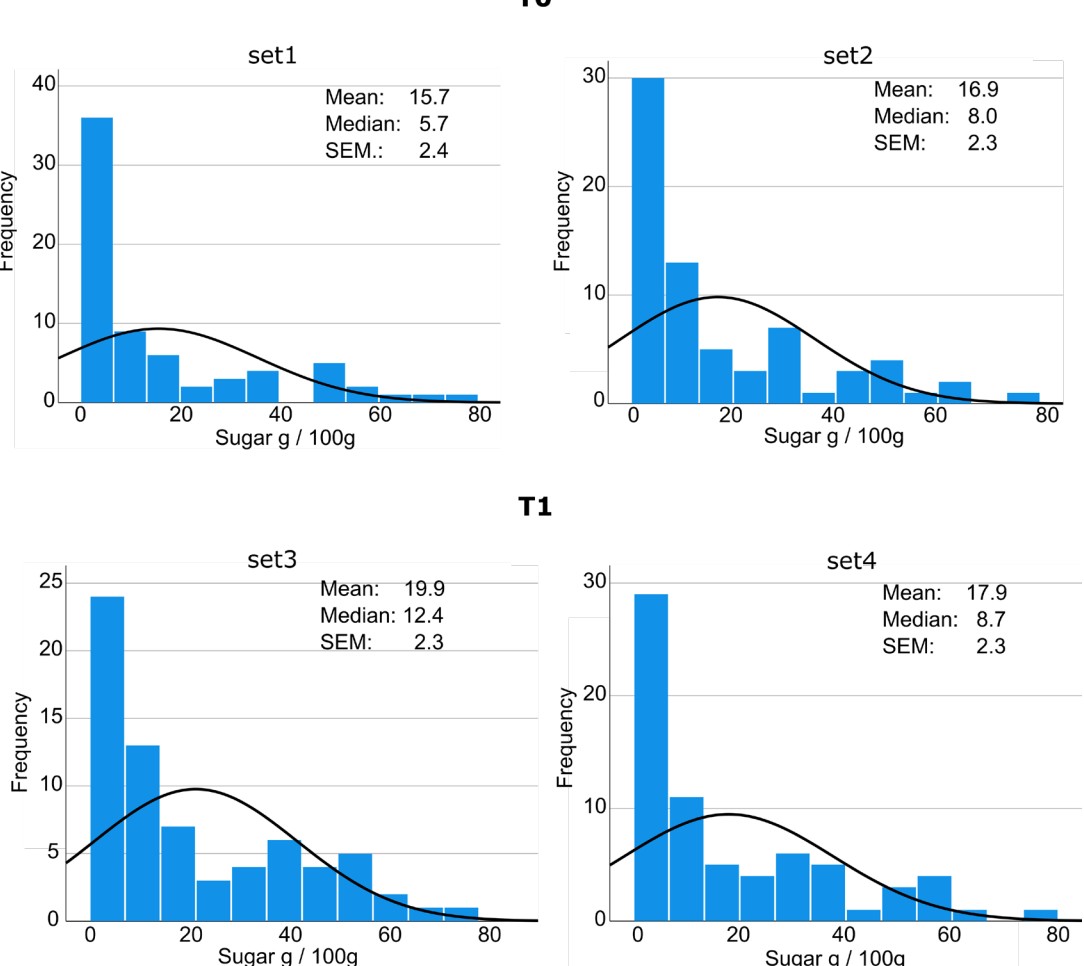

**Appendix 1—figure 5.** Distribution of sugar content in stimuli sets. Sugar content was assessed using the food database on fddb.info and did not differ between sets (p > 0.61, analysis of variance [ANOVA]).

