## [Editor Report]

This is a strong translationally relevant study on the importance of insulin and the mesolimbic response to feeding and attempts at weight loss. It will be of great interest to not only neuroscientists but those who study metabolism and nutrition.

---

## [Decision Letter]

**Decision letter after peer review:**

Thank you for submitting your article "Insulin sensitivity in mesolimbic pathways predicts and improves with weight loss in older dieters" for consideration by *eLife*. Your article has been reviewed by 1 peer reviewer, and the evaluation has been overseen by a Reviewing Editor and Carlos Isales as the Senior Editor. The reviewer has opted to remain anonymous.

Essential revisions:

Abstract

(1) Line 22: Whole body insulin sensitivity was not assessed, please write peripheral (same for line 406).

Results

(2) Line 76 ff: Please indicate the age range, BMI range, and HOMA IR range of participants. Please add the number of men and women in the DG and WG group separately. Was sex equally distributed for the intervention and control groups? Were the women in menopause?

(3) The authors present the nucleus accumbens and VTA BOLD response to insulin. As this is quite a novel study, I would recommend showing the whole brain response (uncorrected for display) for Insulin versus placebo for T0 and T1 for both groups (for sweet versus non-sweet contrast). This could be presented in the supplementary material.

Discussion

(4) Line 302: the first statement of the discussion is quite strong. I agree that the regression model shows that peripheral insulin sensitivity and central insulin response explained variance independent of one another, this is a first indication that peripheral and central responses might be independent, but the study also shows that they correlate. Furthermore, insulin resistance was only based on fasting measurements and not during a clamp or oGTT.

Methods

(5) Participants: in the abstract and in the discussion (line 392), the authors refer to their study sample as prediabetic. However, based on the description of the participants in the methods section, persons overweight and with obesity at the age of 55 and higher were recruited for the study. No official criteria for prediabetes (as described for example by the ADA: https://www.diabetes.org/diabetes/a1c/diagnosis) were employed. Please use the term non-diabetic as in other passages of the manuscript.

(6) I agree that C-peptide-based assessments of peripheral insulin resistance can be superior to insulin-based calculations. However, the CPR-IR is rarely used (only 9 citations of the original paper on that index). I suggest using the commonly used C-peptide-based HOMA-2 instead (DOI: 10.2337/diacare.21.12.2191 see: https://www.dtu.ox.ac.uk/homacalculator/index.php).

Why did the authors use plasma glucose of 128 mg/dl as a cut-off for non-fasting? A commonly used cut-off is 126 mg/dl. Another reliable way to detect non-fasted individuals is unusually high C-peptide concentrations.

(7) Line 420: "two showed task behavior indicating...". I think there is a word missing.

(8) Line 422: "substantial movement". How was this defined? This could also be indicated in paragraph 522 ff.

(9) The authors conducted a median split on sugar content to divide the stimulus set into a high sugar (HS) and low sugar (LS) condition. What was the distribution of the sugar content in the food images? Who evaluated the sugar content of the stimuli?

(10) Line 433: The exclusion criteria include certain illnesses. Was this assessed by a physician or self-reported? Please add this information.

(11) Line 440 ff: What is meant by "additional financial incentives were provided [..] for successful weight loss"? Did participants receive more money if they lost more weight? Please clarify.

(12) Line 465: How was the individual reduction by 10 to 15% calculated? Please specify. Was resting energy expenditure used? Or was it based on their daily prior consumption? The authors mention dietary records. How many days were recorded and at which time points of the study? How was compliance assured?

(13) Line 516: it is not clear what 30_ steeper means. Please clarify.

(14) Line 502: The fMRI food cue paradigm relies on images selected from the internet. Please indicate from what platform these images were selected. The images were then categorized based on sweetness. Were the pictures matched for physical and psychological variables? The authors only mention that the mean preference ratings of the stimuli were investigated in an independent sample, but other properties could play an important role (e.g. arousal, valence, and complexity of image).

(15) The fMRI data analyses section is described in detail up to the second level model.

Line 540: For the second level model, the authors indicate that they used one-sample t-tests and two-sample t-tests. Please be more specific about which contrasts (differential images) went into the model. (16) Based on the described results, I would have expected paired t-tests (T0 versus T1) or a flexible factorial model to investigate within-subject and between-subject effects in one model. Please clarify.

(16) The method section is lacking a statistical methods paragraph for the regression analysis and behavioral data analyses. What tool was used, etc.?

(17) The authors used one-way ANOVAs for analyzing parametric liking scores for the different picture sets. What is the distribution of the liking ratings? Are they normally distributed? The authors computed several rmANOVAs. Were the assumptions for such analyses tested?

BMI reduction was correlated with several measures (predictors of subsequent weight loss). In this (18) case, you must correct your p-values in relation to the number of tests (Bonferroni correction). Were the correlations between the BOLD response and peripheral insulin resistance still significant after adjusting for BMI and age?

(19) Was the study registered, for example at clinical trials.gov? Were the hypothesis and primary outcomes or study design preregistered? How was weight loss success defined?

(20) The study by Tiedmann et al., 2017 Nat Comm used the same fMRI paradigm to investigate insulin response of the reward circuitry in young adults of normal weight as well as overweight and obesity. In that study, the authors used HOMA-IR as a measure of peripheral insulin resistance. Why change this in the current study? Furthermore, in the previous study by Tiedemann, DCM model was used to show insulin-induced changes in connectivity between the nucleus accumbens and VTA. This connection was significantly modulated by intranasal insulin. Why wasn't this evaluated in the current study?

(21) The study of Tiedemann published in 2017 investigated the response of intranasal insulin in young individuals, while the current study included elderly persons. The impact of age independent of BMI has not been investigated thus far on neural insulin processing. Have you looked at the effect of age on the nucleus accumbens or VTA BOLD response to insulin?

(22) The authors use the term insulinergic inhibition. Is it possible to evaluate neural inhibition with BOLD fMRI?

---

## [Author Response]

Essential revisions:Abstract(1) Line 22: Whole body insulin sensitivity was not assessed, please write peripheral (same for line 406).

This has been changed.

Results(2) Line 76 ff: Please indicate the age range, BMI range, and HOMA IR range of participants. Please add the number of men and women in the DG and WG group separately. Was sex equally distributed for the intervention and control groups? Were the women in menopause?

We have now added age range, BMI range, and HOMA-2 (+range). Number of men and women in DG and WG were also added. We also mention that there were no sex differences between groups (chi-square = 1.39, P = .38).

We did not specifically assess menopause status by hormonal measurements. The youngest woman was 55 years old, so it is likely that most, if not all, women in this study were in the middle or postmenopausal phase.

(3) The authors present the nucleus accumbens and VTA BOLD response to insulin. As this is quite a novel study, I would recommend showing the whole brain response (uncorrected for display) for Insulin versus placebo for T0 and T1 for both groups (for sweet versus non-sweet contrast). This could be presented in the supplementary material.

We agree that sub-threshold activation patterns are helpful to inform future designs given the novelty of our study. Consistent with the reviewer’s suggestion, we have therefore included uncorrected SPMs for each time point for all participants in the appendix (Appendix Figure 3). In addition, we would like to point out that whole-brain T-maps of these and the other results are available at Dryad (https://datadryad.org/stash/share/_sUG1RvcWB3uCpSxeiYRWcKBtUZmCnzSjir4dWGv16c) which can be used, for example, to suggest neural assumptions in future studies.

Discussion(4) Line 302: the first statement of the discussion is quite strong. I agree that the regression model shows that peripheral insulin sensitivity and central insulin response explained variance independent of one another, this is a first indication that peripheral and central responses might be independent, but the study also shows that they correlate. Furthermore, insulin resistance was only based on fasting measurements and not during a clamp or oGTT.

We have toned down this statement in accordance with the reviewer’s suggestion in “Our findings indicate…”. However, we would like to point out that the regression model indeed shows independent effects of the two parameters (based on partial correlations) and that the two variables indeed were not correlated at T0. Both aspects strongly suggest that we observed two independent predictive mechanisms in this context.

We agree that an OGTT or clamp based assessment of insulin sensitivity would have been an interesting addition to this study When we designed the study, we decided against an euglycemic clamp procedure or oGTT, mainly for logistic and compliance reasons. The latter, in particular, could have had an impact on participants’ performance in the scanner. Participants already had to fast overnight for at least 10 hours, meaning that scanning took place in the early morning. Considering that one study day already lasted 2 hours (before participants were finally allowed to eat), a preceding clamp or oGTT would have probably overstretched the physical and psychological compliance of some participants. Reassuringly, there is an extensive literature demonstrating the high validity (e.g., reported correlations of HOMA-IR with clamp assessment ~.88) of fasting measurements in non-diabetic participants (reviewed in Wallace et al., Diabetes Care, 2004).

Methods(5) Participants: in the abstract and in the discussion (line 392), the authors refer to their study sample as prediabetic. However, based on the description of the participants in the methods section, persons overweight and with obesity at the age of 55 and higher were recruited for the study. No official criteria for prediabetes (as described for example by the ADA: https://www.diabetes.org/diabetes/a1c/diagnosis) were employed. Please use the term non-diabetic as in other passages of the manuscript.

This has been changed.

(6) I agree that C-peptide-based assessments of peripheral insulin resistance can be superior to insulin-based calculations. However, the CPR-IR is rarely used (only 9 citations of the original paper on that index). I suggest using the commonly used C-peptide-based HOMA-2 instead (DOI: 10.2337/diacare.21.12.2191 see: https://www.dtu.ox.ac.uk/homacalculator/index.php).

In accordance with this suggestion, we now report C-peptide-based HOMA-2 values. This resulted in minor changes in the results (HOMA-2 contains a non-linear transformation that altered the ranking of the T0-T1 difference values) but did not affect the main findings of the study. CPR-IR and HOMA-IR values were removed.

Why did the authors use plasma glucose of 128 mg/dl as a cut-off for non-fasting? A commonly used cut-off is 126 mg/dl. Another reliable way to detect non-fasted individuals is unusually high C-peptide concentrations.

Please excuse that this phrase was misleading. Our cut-off was actually 126, but what was meant was that all three excluded individuals had a value of > 128 (i.e., 129, 131, 134). In the remaining sample, there was no glucose value higher than 125 on any of the four study days and we have now included this information in the Methods section: Three participants were excluded because of elevated glucose levels (≥126)

(7) Line 420: "two showed task behavior indicating...". I think there is a word missing.

This has been corrected.

(8) Line 422: "substantial movement". How was this defined? This could also be indicated in paragraph 522 ff.

The following information was added to the suggested paragraph: Image diagnostics was performed using visual inspection of image-to-image variability (tsdiffana, https://imaging.mrc-cbu.cam.ac.uk/imaging/DataDiagnostics).

This visualization revealed massive distortions in one participant that matched the logged observations from the scanning sessions. The reason for these artifacts was most likely due to the pronounced head circumference of this very large participant, which prevented him from being optimally positioned in the head coil.

(9) The authors conducted a median split on sugar content to divide the stimulus set into a high sugar (HS) and low sugar (LS) condition. What was the distribution of the sugar content in the food images? Who evaluated the sugar content of the stimuli?

The sugar content in all four stimulus sets was evaluated using the food database on fddb.info. Importantly, the four parallel stimulus sets did not differ significantly w.r.t. mean and median sugar content. This is particularly important for the sets used at placebo and insulin at T0 and T1 respectively. The use of the first two sets was counterbalanced for IN/PL at T0, the third and fourth sets were counterbalanced for IN/PL at T1. Information on the distribution of the sugar content for each set has now been added to the Methods and the Appendix (Appendix Figure 5).

(10) Line 433: The exclusion criteria include certain illnesses. Was this assessed by a physician or self-reported? Please add this information.

Initial screening as well as all clinical measurements in this study were performed by a physician (P.F., K.G.). This information was added in lines 433 ff.

(11) Line 440 ff: What is meant by "additional financial incentives were provided [..] for successful weight loss"? Did participants receive more money if they lost more weight? Please clarify.

Participants received an extra bonus of 50€. We have now added this information.

(12) Line 465: How was the individual reduction by 10 to 15% calculated? Please specify. Was resting energy expenditure used? Or was it based on their daily prior consumption? The authors mention dietary records. How many days were recorded and at which time points of the study? How was compliance assured?

The individual reduction was based on participants’ daily prior consumption as assessed by dietary records. For these records, participants protocolled their daily food intake for one week (7 days).

Compliance was assured by three telephone contacts after 2, 6 and 12 weeks during which participants confirmed that they continued to adhere to the diet and had the opportunity to clarify any questions regarding the dietary regimen. We have added this information.

(13) Line 516: it is not clear what 30_ steeper means. Please clarify.

Thank you, it should be 30° steeper, and has been corrected.

(14) Line 502: The fMRI food cue paradigm relies on images selected from the internet. Please indicate from what platform these images were selected. The images were then categorized based on sweetness. Were the pictures matched for physical and psychological variables? The authors only mention that the mean preference ratings of the stimuli were investigated in an independent sample, but other properties could play an important role (e.g. arousal, valence, and complexity of image).

Stimuli were not selected from a particular platform. They were selected via a Google search from a predefined list that we had created to ensure that a similar amount of savory and sweet foods were included. Sugar in all four stimulus sets was evaluated using the food database on fddb.info. Sugar content did not differ between sets. The correspondence of “valence” between the four stimulus sets was previously validated in an independent sample of 16 participants using liking ratings. We have now made a further subdivision into HS and LS stimuli and, as a kind of approximation to “arousal” / visual complexity, we have done the same for picture saliency (line 516 ff). The data are presented in the appendix (Appendix Table 2), and show no difference between stimulus categories and the sets.

(15) The fMRI data analyses section is described in detail up to the second level model.Line 540: For the second level model, the authors indicate that they used one-sample t-tests and two-sample t-tests. Please be more specific about which contrasts (differential images) went into the model.

The following was added to the Methods section: Contrast images of interest comprised onset and parametric regressors for food > non-food and HS > LS from the placebo session at T0 (general paradigm-induced activation) and parametric regressors for HS > LS covering insulin effects at baseline (PL > IN) and compared to follow-up (T1 > T0).

In addition, we made sure that all contrasts that went into second level summary statistics are now specified in the result section and the figure legends.

(16) Based on the described results, I would have expected paired t-tests (T0 versus T1) or a flexible factorial model to investigate within-subject and between-subject effects in one model. Please clarify.

We preferred the more conservative summary statistics (T-Tests) to factorial designs primarily for the following reasons: Mixed-effects models (i.e., with both within-subjects and between-subject factors, as in our study) require full variance partitioning. Repeated measures ANOVAs (i.e., flexible designs) in SPM provide only one variance term, in this case, within-subject variance. This means that the results of group main effects as well as individual condition main effects can be inflated and therefore require separate models (for more details on this topic, see McFarquhar: Modeling group-level repeated measurements of neuroimaging data using the univariate general linear model. Frontiers in Neuroscience, 13, 2019). Summary statistics are robust to potential misspecification, and since we were primarily interested in analyses with covariates (i.e., simple regressions), we set up all the relevant contrasts at the single subject level and submitted each of the contrast images in a separate one- or two-sample T-test which corresponds to results from an ANOVA with partitioned error terms.

(16) The method section is lacking a statistical methods paragraph for the regression analysis and behavioral data analyses. What tool was used, etc.?

Thank you, this paragraph has been added.

(17) The authors used one-way ANOVAs for analyzing parametric liking scores for the different picture sets. What is the distribution of the liking ratings? Are they normally distributed? The authors computed several rmANOVAs. Were the assumptions for such analyses tested?

There are three main assumptions for rmANOVAs:

1) Independent observations: Given for individual subjects.

2) Sphericity of all difference scores among test variables: We used only two-level within subject factors (i.e., pre vs. post, PL vs. IN, HS vs. LS), that is, testing for sphericity of all differences (e.g., using Mauchly’s Test) is not applicable. In other words, there is no possible violation of this assumption.

3) Normality: We now have tested for normality for all variables of interest both across participants and within groups using the Kolmorgorov-Smirnov-Test (line 502 ff). The main results here are:

HS and LS liking scores before and after the scan, at T0 and T1, are normally distributed across participants and within groups (all P >.098).

HOMA-2 scores at T0 and T1 and difference scores across and within groups are normally distributed (all P >.20)

We also tested all other variables included in control analyses (hunger, time fasted, glucose, insulin, c-peptides). Here, we found at least one single measure (from pre or post, PL or IN, T0 or T1) was not normally distributed for fasting time, hunger and glucose. Nonparametric tests are now given for these variables (Table 1, Methods line 503 ff, Appendix Tables 1 and 5). As expected, the results did not differ from those of the parametric testing.

BMI reduction was correlated with several measures (predictors of subsequent weight loss). In this (18) case, you must correct your p-values in relation to the number of tests (Bonferroni correction).

We have now restricted the reported correlations to those related to our main hypotheses on weight loss prediction, i.e., one correlation with peripheral insulin sensitivity and one correlation with neural insulin sensitivity. The neural results are already corrected by family wise error (FWE) correction. In addition, the multiple regression model was used to test the predictive value of both parameters within a single model (partial correlations). To address the reviewer’s concern, however, we Bonferroni corrected the regression coefficients. Both coefficients survived the corrected threshold in our model of interest. In the control analysis with BMI as a third predictor, peripheral insulin sensitivity survived only an uncorrected threshold.

Were the correlations between the BOLD response and peripheral insulin resistance still significant after adjusting for BMI and age?

As suggested by the Reviewer, we performed a control analysis including BMI and age as covariates in the regression model of central and peripheral insulin changes. This analysis again revealed significant results in the left NAc with an identical peak voxel (-10, 10, -7, P <.05, FWE corrected). This information was added to the results. For standardization, we also performed this control analysis for baseline findings in the VTA. Again, the addition of the covariates age and BMI did not result in a difference for the significant BOLD response in the VTA.

(19) Was the study registered, for example at clinical trials.gov? Were the hypothesis and primary outcomes or study design preregistered? How was weight loss success defined?

Unfortunately, when the study was initiated in 2015, we did not pre-register the project (as we do by default today). This study is based on a peer-reviewed grant proposal where hypotheses are specified and that can be provided by request. It is now post-hoc –registered, but we admit that this is not very satisfactory. Basically, the project was not designed as a clinical trial with no primary focus on clinical outcomes but as a basic research study of how peripheral and central insulin function relates to weight changes in this particular population. For this reason, weight loss (outcome) was only defined by BMI changes.

(20) The study by Tiedmann et al., 2017 Nat Comm used the same fMRI paradigm to investigate insulin response of the reward circuitry in young adults of normal weight as well as overweight and obesity. In that study, the authors used HOMA-IR as a measure of peripheral insulin resistance. Why change this in the current study?

The reason for using a c-peptide based measure was that in the current study we observed a surprisingly high within-subject variability in insulin compared to c-peptide levels in some participants. This was also statistically relevant, as evidenced by significantly higher coefficients of variance compared with c-peptide levels. Due to this variability together with reports that c-peptide based indices are a more reliable indicator of insulin secretion, we preferred a c-peptide based measure and are grateful for the reviewer’s suggestion to use HOMA-2 as a further updated model of HOMA-IR.

Furthermore, in the previous study by Tiedemann, DCM model was used to show insulin-induced changes in connectivity between the nucleus accumbens and VTA. This connection was significantly modulated by intranasal insulin. Why wasn't this evaluated in the current study?

In Tiedemann et al., 2017, we were interested in whether consistent animal findings on insulin action in mesolimbic networks / connections can be translated into the human brain and our DCM results mainly confirmed this translation. In the current study, we used these results to define regions of interest for studying individual predictors and consequences of weight changes. In fact, we had no hypotheses for a generally different basic insulinergic mechanism in this particular sample. But if there are hypotheses of age-related changes in basic insulin function, this should first be tested in healthy, lean older (compared with younger) adults, perhaps involving structural measures. We agree that this could be an interesting question to address in the future. However, the focus of the current project was on longitudinal aspects of the interactions between insulin and weight, and the amount of data and analyses reported here are already quite complex.

(21) The study of Tiedemann published in 2017 investigated the response of intranasal insulin in young individuals, while the current study included elderly persons. The impact of age independent of BMI has not been investigated thus far on neural insulin processing. Have you looked at the effect of age on the nucleus accumbens or VTA BOLD response to insulin?

We agree that age effects per se are very interesting. At the moment, we do not have data from healthy, lean, older adults to examine this aspect, but we hope to address this question in an upcoming project. At least, as suggested by the reviewer and addressed in response #18, our results hold when including age and BMI as covariates.

(22) The authors use the term insulinergic inhibition. Is it possible to evaluate neural inhibition with BOLD fMRI?

The term insulinergic inhibition is based primarily on our neural and behavioral findings showing (a) reduced food preference, and (b) reduced BOLD signal under insulin compared to placebo. This fits strikingly with data in rodents showing insulinergic inhibition of neural activity in eating-relevant mesolimbic systems (e.g., Stuber and Wise, Nature Neuroscience 2016). Of course, we are unable to differentiate direct or indirect inhibitory effects of insulin, but we believe that our current results, together with our previous data and animal findings, support this “inhibitory” interpretation.